# Multiple Direct Effects of the Dietary Protoalkaloid *N*-Methyltyramine in Human Adipocytes

**DOI:** 10.3390/nu14153118

**Published:** 2022-07-29

**Authors:** Christian Carpéné, Pénélope Viana, Jessica Fontaine, Henrik Laurell, Jean-Louis Grolleau

**Affiliations:** 1Institut des Maladies Métaboliques et Cardiovasculaires (I2MC), Institut National de la Santé et de la Recherche Médicale (INSERM U1297), I2MC, CEDEX 4, 31432 Toulouse, France; penelope.viana@inserm.fr (P.V.); jessica.fontaine@inserm.fr (J.F.); henrik.laurell@inserm.fr (H.L.); 2CHU Rangueil, Université Paul Sabatier, I2MC, CEDEX 4, 31432 Toulouse, France; 3Department of Plastic Surgery, CHU Rangueil, 31059 Toulouse, France; grolleau.jl@chu-toulouse.fr

**Keywords:** adipocytes, insulin-stimulated glucose transport, dietary amines, monoamine oxidase, semicarbazide-sensitive amine oxidase, triacylglycerol breakdown, obesity

## Abstract

Dietary amines have been the subject of a novel interest in nutrition since the discovery of trace amine-associated receptors (TAARs), especially TAAR-1, which recognizes tyramine, phenethylamine, tryptamine, octopamine, *N*-methyltyramine (NMT), synephrine, amphetamine and related derivatives. Alongside the psychostimulant properties of TAAR-1 ligands, it is their ephedrine-like action on weight loss that drives their current consumption via dietary supplements advertised for ‘fat-burning’ properties. Among these trace amines, tyramine has recently been described, at high doses, to exhibit an antilipolytic action and activation of glucose transport in human adipocytes, i.e., effects that are facilitating lipid storage rather than mobilization. Because of its close structural similarity to tyramine, NMT actions on human adipocytes therefore must to be reevaluated. To this aim, we studied the lipolytic and antilipolytic properties of NMT together with its interplay with insulin stimulation of glucose transport along with amine oxidase activities in adipose cells obtained from women undergoing abdominal surgery. NMT activated 2-deoxyglucose uptake when incubated with freshly isolated adipocytes at 0.01–1 mM, reaching one-third of the maximal stimulation by insulin. However, when combined with insulin, NMT limited by half the action of the lipogenic hormone on glucose transport. The NMT-induced stimulation of hexose uptake was sensitive to inhibitors of monoamine oxidases (MAO) and of semicarbazide-sensitive amine oxidase (SSAO), as was the case for tyramine and benzylamine. All three amines inhibited isoprenaline-induced lipolysis to a greater extent than insulin, while they were poorly lipolytic on their own. All three amines—but not isoprenaline—interacted with MAO or SSAO. Due to these multiple effects on human adipocytes, NMT cannot be considered as a direct lipolytic agent, potentially able to improve lipid mobilization and fat oxidation in consumers of NMT-containing dietary supplements.

## 1. Introduction

Amines present in mammalian tissues at very low (nanomolar) concentrations, such as tyramine, tryptamine, octopamine and related phenethylamines, are often referred to as “trace amines”. Since many of these amines can occur at higher concentrations than traces in numerous edible plants, foods and beverages (e.g., 2 mg/g of tyramine in fermented cheeses), the term ‘dietary amines’ will be preferred in the current report. Among these dietary amines, *N*-methyltyramine (NMT) is a methylated derivative of tyramine, a small molecule consisting of an aromatic ring branched with a CH_2_-CH_2_-NH_2_ amine moiety. Tyramine is the prototype of indirectly acting sympathomimetic amines, owing almost all its activity to the release of endogenous noradrenaline but also to direct activation of trace amine-associated receptors (TAARs) (for review, see [1,2]). These receptors, especially the TAAR-1 subtype, are widely distributed in brain and peripheral tissues, including in humans [3], known to be involved in the regulation of pancreatic islet hormone secretion [4] and in the modulation of food intake [5]. While *p*-tyramine can be synthesized in the body by decarboxylation of the amino acid L-tyrosine, its daily intake is largely dependent on diet composition as well as on gut microbiota. A similar duality of endogenous and nutritional sources also applies for other dietary amines. However, excessive tyramine intake is sadly known for the ‘cheese effect’ it might trigger, which is a dangerous hypertensive crisis increasing the risk of intracranial hemorrhage occurring in patients treated with inhibitors of monoamine oxidases (MAO) [1,6]. In healthy subjects, tyramine is also known to provoke a hypertensive response, but the amount of tyramine required to increase blood pressure is much higher when ingested than when injected, and even higher when diluted in a complete meal [1,7]. In contrast, tyramine has been reported to directly exert a vasorelaxant effect on rat aorta, which was endothelium independent and not exerted via adrenergic receptor stimulation [8]. Then, this effect could be explained by a probable activation of endothelium-independent mechanisms involving hydrogen peroxide and subsequent activation of soluble guanylyl cyclase [9]. Alongside its cardiovascular duality, tyramine has also been found to exhibit dual actions on the management of fat storage: lipolytic when administered via microdialysis probe in the subcutaneous human adipose tissue (hAT), and inactive when incubated with isolated adipocytes [10]. These differential effects were explained by the capacity of tyramine to release norepinephrine in hAT and by opposite presynaptic and postsynaptic mechanisms. More recently, it has been observed that tyramine inhibits lipolysis [11] and activates glucose transport [12] in human adipocytes, when present at 0.01–1 mM during in vitro experiments lasting less than two hours. These direct effects were dependent on oxidation by monoamine oxidases, and mediated by hydrogen peroxide. More importantly, these effects resemble the actions of insulin on fat cells, namely its antilipolytic and lipogenic components.

Contrarily to the recommendations aiming at limiting tyramine intake to prevent the risk of cardiovascular complications or migraine crises, the intake of its *N*-methylated derivative NMT is encouraged by diverse advertisements of nutritional supplements marketed for lipid mobilization, fat burning and weight loss.

In humans, the enzyme, phenylethanolamine *N*-methyltransferase, found primarily in the adrenal medulla and implied in the conversion of noradrenaline (norepinephrine) to adrenaline (epinephrine), is involved in the metabolism of *p*-tyramine into endogenous *N*-methyltyramine (NMT). However, there are exogenous sources of NMT, since this amine is a protoalkaloid naturally present in plants and it is ingested with diverse foods and beverages, including *Citrus* fruits [13], germinated barley and beer, in which its concentration varies between 0.5 and 4.5 mg/L [14]. NMT intake releases gastrin and gastric secretion [15]. However, NMT daily consumption is supposed to be low. Thus, NMT belongs to ingredients that are present in—or added to—dietary supplements commonly used for weight loss and athletic performance, also containing related phenetylamines such as synephrine and octopamine. Such supplements have been marketed once the use of ephedrine was banned as nutritional anti-obesity approach. Previous studies indicated that NMT is a weak activator of lipolysis in rodents and in humans [16,17], but, to our knowledge, its capacity to reproduce or counteract insulin action on glucose metabolism in fat cells has never been documented.

The following results will show similarities between the in vitro actions of tyramine and NMT, regarding their partial activation of glucose transport, weak activation of lipolysis, impairment of β-adrenergic activation of triacylglycerol breakdown, and interplay with amine oxidases in adipocytes. These similarities do not justify why tyramine is almost banned in the diet of patients treated with MAO irreversible inhibitors or suffering from migraine crises, while NMT is advertised to promote body weight loss and lipid mobilization.

## 2. Materials and Methods

### 2.1. Human Adipose Tisue Samples

The adipose samples were obtained from a total of 76 women undergoing abdominal surgery at the Rangueil hospital, Toulouse (France) with their informed consent. Their mean body mass index (BMI) was 25.9 ± 0.3 kg/m^2^ (age range: 25–70 year). This study was validated by the local ethics committee for the protection of individuals (Comité de Protection des Personnes Sud Ouest et Outre-Mer II) under the reference: DC-2014–2039. For studies on glucose transport in isolated adipocytes, the cells needed to be freshly isolated from the abdominal adipose tissue samples and to be handled immediately after surgical intervention. Thus, following surgical removal in the operating room, the pieces of subcutaneous abdominal human adipose tissue (hAT) were transported to the laboratory in less than half an hour and the adipocytes were—after tissue digestion and subsequent washings—distributed in Eppendorf plastic tubes for glucose uptake assays in less than three hours. All the procedures were performed at 37 °C. Small portions (approximately 2 g) of hAT were not subjected to collagenase digestion and snap frozen, then stored at −80 °C until determination of amine oxidase activity on thawed preparations.

### 2.2. Adipocyte Preparation, Glucose Transport and Lipolysis Assays

Adipocytes were isolated from minced fat tissue under agitation by using collagenase at 1–3 mg/mL as already detailed [18]. The medium used for the collagenase digestion of hAT was Krebs–Ringer containing 15 mM sodium bicarbonate, 10 mM HEPES, 3.5% bovine serum albumin (pH 7.4). It was lacking of glucose to allow the tracking of the non-metabolizable hexose analogue 2-[1, 2-^3^H]-D-deoxyglucose (2-DG). To supply energy to the cells during the in vitro experiments, this medium contained 2 mM pyruvate, as described in [19]. The 2-DG uptake was stopped by the addition of 100 µM cytochalasin B, and by a separation between intracellular and extracellular hexose that was performed as previously described [12], via a spin through a dynonylphtalate layer in order to obtain undamaged radiolabeled fat cells [19].

In 16 individuals, lipolytic activity was assessed by determining the glycerol released from adipocytes incubated with the tested agents during 90 min as described in [11]. The response to 10 µM isoprenaline was set as 100% maximal lipolytic activity. In these conditions, the release of free fatty acids has been already demonstrated to follow the pattern of glycerol release [17]. To better detect antilipolytic responses, submaximal stimulation elicited by 100 nM isoprenaline was applied to adipocytes simultaneously with the addition of tested agents, and the remaining glycerol release that resisted to antilipolytic agent was expressed as the percentage of maximal lipolytic activity.

### 2.3. Tyramine and Benzylamine Oxidation by Human Adipose Tissue

The oxidation of 0.5 mM [^14^C]-tyramine and 0.5 mM [^14^C]-benzylamine was performed as described in [20], using hAT homogenates as biological source of amine oxidases. Frozen pieces of subcutaneous abdominal adipose depots were thawed and homogenized in 200 mM phosphate buffer (pH 7.5) for 30 s followed by 10 s sonication as described in [20]. The preparation of homogenates was performed at room temperature to avoid the formation of a fat cake, which becomes solid at colder temperatures and could alter sample distribution, as detailed in [21]. For a total of 14 individuals, the protein amount distributed in each assay tube of 200 µL averaged 117 ± 8 µg protein.

Amine oxidation was then determined on 30-min incubation at 37 °C, with the competing agents added without pre-incubation at time 0, simultaneously with the radiolabeled substrates, as already described [22]. At the end of the incubation, the neo-synthesized products of amine oxidation, (mainly mandelic acid and benzaldehyde) were extracted in ethyle acetate/toluene and counted in liquid scintillation cocktail, as in [23]. The oxidation of 0.5 mM substrate without any competing agent represented 1.5 and 6.0 nmol/mg protein/min for [^14^C]-tyramine and [^14^C]-benzylamine, respectively, and was set at 100% reference. In these conditions, the MAO inhibitor pargyline impaired more than 95% of tyramine oxidation, while SSAO inhibitors (aminoguanidine or BTT 2052) abolished at 1 mM all the production of benzylamine-related radioactive aldehydes.

### 2.4. Reagents

2-[1, 2-^3^H]-D-deoxyglucose, [7-^14^C]-benzylamine and the liquid scintillation cocktail Emulsifier-Safe were from Perkin Elmer SAS (Waltham, MA, USA). [^14^C]-tyramine was obtained from Sigma-Aldrich-Merck (Saint Quentin Fallavier, France). *N*-methyltyramine, *p*-tyramine, pargyline, synephrine, bovine albumin, collagenase and most of the other chemicals were from Sigma-Aldrich-Merck (Saint Quentin Fallavier, France). The indane hydrazino alcohol BTT 2052 was a generous gift from D. Smith and M. Salmi (Turku, Finland). The 2-benzofuranyl-2-imidazoline (BFI) was kindly given by Dr. A. Hudson (Univ. of Alberta, Edmonton, AB, Canada).

### 2.5. Statistical Analyses

Results are given as the means ± SEM of n observations. Statistical analyses used ANOVA followed by post hoc Dunnett’s test for multiple comparisons, performed with Prism 6 for Mac OS X (from GraphPad software, Inc., San Diego, CA, USA). When indicated, Student’s paired *t* tests were used. NS means no significant difference between compared samples. ND means not determined.

## 3. Results

### 3.1. N-Methyltyramine Reproduces the Stimulatory Effect of Tyramine on Glucose Uptake in Human Adipocytes

Since tyramine has been reported to activate glucose transport in human adipocytes [12], it was verified whether *N*-methyltyramine (NMT) could share the same property. Figure 1 summarizes this first comparative approach. On a total of 52 individuals who underwent abdominal plastic surgery in the period 2014–2020, a highly significant stimulatory action of 1 mM tyramine was found when compared to spontaneous 2-DG uptake (basal) (Figure 1a). The data representation chosen for Student’s *t* paired analysis used in the tyramine panel also illustrated the substantial inter-individual variability that exists in the baseline of hexose uptake in human adipocytes. Data were redrawn from results obtained on freshly isolated adipocytes during our previous in vitro determinations of 2-deoxyglucose uptake (2-DG) [12]. A similar comparison between basal uptake and response to 1 mM NMT was performed in other adipocyte preparations obtained from 24 patients examined between 2018 and 2021 (Figure 1b). A highly significant stimulatory action was observed for 1 mM NMT. This comparative approach indicated that both amines induced short-term activation of glucose transport in human adipocytes. Given the similarities between their chemical structure and effect, NMT was supposed to act as already stated for tyramine [12], i.e., as a MAO substrate generating via oxidative deamination a sufficient amount of hydrogen peroxide to trigger the translocation of glucose transporters to the fat cell surface.

### 3.2. Dose Dependency of Glucose Uptake Activation by NMT and Related Phenethylamines

The comparison of the stimulatory effects of NMT and tyramine was then extended to other biogenic amines and to insulin, the reference activator of glucose transport.

In 8 preparations of human adipocytes, the maximal activation of hexose uptake by insulin was equivalent to a 4.1 ± 0.5 fold increase over baseline (Figure 2). Confirming our previous findings, millimolar doses of tyramine [12], benzylamine [24] and methylamine [25] partially reproduced the insulin stimulation of 2-DG uptake. An increase in glucose uptake was also detected with 0.01 mM NMT, while a plateau was reached between 0.1 and 1 mM NMT (Figure 2). This response to sub-millimolar doses of NMT was limited to approximately one-third of the stimulation induced by 100 nM insulin and did not differ significantly from the responses to known amine oxidase substrates, i.e., benzylamine, methylamine and tyramine (Figure 2).

Three other amines were also tested at 0.1 and 1 mM under the same conditions: (1) *p*-synephrine, a natural alkaloid chemically close from NMT since it differs only by a single additional hydroxyl group; (2) prenylamine, an old drug of the amphetamine family, which is less structurally related since it contains three aromatic cycles, and which has been prescribed for its vasodilator and cardiodepressant properties, then withdrawn from the market for toxicity reasons; (3) glucosamine, which is chemically distinct since it is an aminosugar that contributes to the processes of glycation and of enzymatic glycosylation, such as in the case of mucopolysaccharides. Only synephrine activated glucose uptake in human adipocytes to an extent that resembled that of the other active dietary amines (Figure 2).

Thus, not all the amines activated glucose transport in human adipocytes. The results also suggested that the mechanisms involved in the capacity of any given amine to stimulate glucose transport are not solely related to its ability to interact with monoaminergic systems.

### 3.3. Sensitivity of NMT Activation of Hexose Uptake to Amine Oxidase Inhibitors

The common mechanism of action depicted for the three amines of reference (tyramine, benzylamine, methylamine) is related with amine oxidase-dependent production of hydrogen peroxide and is relatively independent from interplay with other amine transporters, monoaminergic receptors or metabolizing enzymes [12,24]. We therefore investigated whether the NMT effect on hexose uptake was impaired by amine oxidase inhibitors. On another set of 8 adipocyte preparations obtained from women undergoing subcutaneous abdominal plastic surgery, the level of 2-DG uptake was 0.33 ± 0.04 nmol/100 mg lipids/10 min for basal and raised up to 1.38 ± 0.12 nmol/100 mg lipids/10 min in response to 100 nM insulin (*p* < 0.001). As above, the effects of 1 mM benzylamine, NMT or tyramine were significant (0.61 ± 0.09, 0.54 ± 0.07 and 0.50 ± 0.07 nmol/100 mg lipids/10 min, respectively, *p* < 0.02) and were tested in the presence of the four following inhibitors: (1) pargyline, a first-generation MAO inhibitor; (2) aminoguanidine, a drug that inhibits both SSAO (also named AOC3 or VAP-1) and diamine oxidase (DAO, also named AOC1) [26]; (3) BTT 2052, an indane hydrazino alcohol that blocks SSAO activity; and (4) 2-(2-benzofuranyl)-2-imidazoline (BFI), an imidazoline derivative demonstrated to inhibit human MAO-A, which is abundantly expressed in human adipose tissue [27]. To circumvent the inter-individual variations of 2-DG uptake in human adipocyte preparations as evidenced in Figure 1, the glucose transport activation by 100 nM insulin was taken as a positive control for the maximal responsiveness and was set at 100% in each of these additional cases, while the basal uptake was set at 0% (Table 1). At 0.1 mM, pargyline did not modify the basal or the benzylamine-stimulated 2-DG uptake, while it impaired the tyramine-induced stimulation. Pargyline exhibited only a tendency to inhibit the response to NMT. Aminoguanidine was without effect on basal uptake, while at 1 mM, it totally abolished the stimulation induced by each of the three amines. The same observations were made for 1 mM BTT 2052. The imidazoline derivative BFI, unable to alter basal 2-DG uptake, blunted the insulin-like effect of NMT as well as that of tyramine (Table 1).

The sensitivity to the tested inhibitors suggested that SSAO activity was predominantly involved in the observed effect of benzylamine, while a contribution of MAO was required for NMT and tyramine actions. Alternatively, it could not be excluded that a loss of selectivity occurred for the SSAO inhibitors used at 1 mM.

### 3.4. Oxidation of Radiolabelled Tyramine and Benzylamine by Adipose Tissue: Competition by NMT and Various Substrates or Inhibitors

Since complementary dose-dependent studies of glucose transport blockade by other selective amine oxidase inhibitors would have required substantial amounts of freshly prepared adipocytes to better define the relative proportion of MAO and SSAO involved in the NMT-induced activation of hexose uptake, we further examined the interaction of NMT with human MAO and SSAO by directly measuring amine oxidase activity. Indeed, the oxidation of amine substrates is catalyzed by one or various amine oxidases present in human adipose tissue (hAT), and no cellular integrity is required for performing inhibition studies of these amine oxidase activities. While such enzymologic approaches are generally carried out with purified enzymes, we used thawed crude preparations of hAT to avoid alterations in the proportion of the various catabolizing enzymes naturally coexisting in this biological material.

As expected, the oxidation of 0.5 mM [^14^C]-tyramine by hAT homogenates was dose dependently inhibited by the MAO inhibitor pargyline (Figure 3a). It was also partially impaired by aminoguanidine. This suggested either that tyramine oxidation by crude hAT preparations was not only dependent on MAO but also on other amine oxidases, or that the selectivity of aminoguanidine to preferentially inhibit copper-containing amine oxidases was lost at higher concentrations. Nevertheless, NMT competed for [^14^C]-tyramine oxidation as well as cold tyramine or serotonin (Figure 3a), indicating that NMT behaved similarly to recognized MAO substrates. On the opposite, isoprenaline, did not compete for [^14^C]-tyramine oxidation (Figure 3a).

When crude hAT preparations were tested for their capacity to oxidize 0.5 mM [^14^C]-benzylamine, the subsequent production of radioactive benzaldehyde was dose-dependently inhibited by aminoguanidine, whereas it remained unaffected in the presence of pargyline. This indicated that SSAO activity was predominantly implied in the oxidation of benzylamine, and was in agreement with previously reported observations [24]. The SSAO substrate methylamine also competed for the oxidation of [^14^C]-benzylamine, while NMT, tyramine and isoprenaline were without notable influence (Figure 3b).

Taken together, these observations suggested that NMT behaved similar to tyramine when subjected to the mixture of amine oxidases coexisting in hAT: both amines appeared as preferential substrates for amine oxidases containing FAD, i.e., MAO-A and MAO-B, rather than for copper-containing amine oxidases, similar to *AOC* gene products and lysyl oxidases. Unfortunately, these competition experiments could not help in better defining the relative proportion of each amine oxidase involved in the metabolism of NMT in hAT. Nevertheless, these results were in line with the sensitivity of NMT-induced hexose uptake to MAO and SSAO inhibition, as observed above on isolated adipocytes.

Again, these findings showed that NMT exerts similar effects than tyramine, at least on hAT amine oxidases. However, it has been described that tyramine exerts in vitro antilipolytic effects in human adipocytes [11], while it can be lipolytic when administered in situ in the subcutaneous hAT of healthy subjects [10]. This prompted us to further compare the lipolytic and antilipolytic responses of human adipocytes to tyramine and NMT, as well as to other reference amines.

### 3.5. Lipolytic and Antilipolytic Effects of NMT in Human Adipocytes

To investigate the stimulatory and inhibitory components of NMT and tyramine on the regulation of triacyglycerol storage in human adipocytes, isoprenaline-stimulated glycerol release was set at 100% for each individual preparation. Figure 4a shows that none of the studied amines was able to totally reproduce the lipolytic effect of the β-adrenergic agonist isoprenaline. Tyramine, benzylamine, methylamine reproduced only 5–25% of the maximal stimulation induced by 10 µM isoprenaline. At 0.1 and 1 mM, NMT was also a poor activator of lipolysis. Except tyramine, all these amines behaved differently from insulin, which did not activate triacylglycerol breakdown (Figure 4a).

In contrast, insulin exhibited an expected antilipolytic action, visualized in Figure 4b by its capacity to reduce the glycerol release induced by isoprenaline (from 90.3 ± 3.2% of maximal glycerol release for 100 nM Iso alone to 64.1 ± 5.8% for 100 nM Iso + 100 nM insulin, n = 16, *p* < 0.01). At 1 mM, the reference amines inhibited glycerol release to a higher extent than insulin itself. At 0.1 mM, NMT fully reproduced the insulin antilipolytic effect. However, at 1 mM, NMT impaired the lipolytic effect of isoprenaline similar to other amines, resulting in approximately 50% inhibition, i.e., greater than that of insulin (Figure 4b).

Taken as a whole, all the adipocyte experiments demonstrated that NMT, tyramine and related amines shared the same ability to partially reproduce the insulin activation of hexose uptake, while they were more efficient than the pancreatic hormone in impairing the stimulation of lipolysis by isoprenaline.

However, in addition to the exploration of the so-called ‘insulin-like actions’ of NMT and tyramine, it was of utmost importance to verify whether these amines could potentiate or hamper the actions of insulin itself. In other words, it was relevant to investigate the potential insulin-sensitizing properties of these dietary amines. This was even more relevant when considering that the dietary amines are ingested together with food or beverages, and potentially reach the adipocyte environment in postprandial periods, characterized by increased insulin levels.

### 3.6. Effect of a Combination of Insulin and NMT on Glycerol Release and Glucose Uptake by Human Adipocytes

To verify whether NMT could amplify or hamper the antilipolytic response to insulin, the influence of amines was studied in the presence of 100 nM isoprenaline and 100 nM insulin in another set of 8 adipocyte preparations. At this dose, isoprenaline induced 88.3 ± 7.0% of maximal lipolysis. This stimulation was lowered to 58.5 ± 8.1% (n = 8, *p* < 0.02) by the addition of 100 nM insulin. Such antilipolytic effect remained unchanged in the presence of 1 µM NMT, which was inefficient on its own (59.0 ± 7.8%, n = 8, NS). The same observation was made when combining insulin and 0.1 mM NMT: 59.6 ± 1.5% (n = 8, NS). When 1 mM NMT was added together with 100 nM insulin to human adipocytes, the remaining stimulation by isoprenaline was lowered to 35.3 ± 5.8% (n = 8, *p* < 0.03), indicating that only a high dose of NMT facilitated the insulin antilipolytic action. However, when considering that the antilipolytic effect of 1 mM NMT lowered on its own the isoprenaline-induced glycerol release to 42.9 ± 9.1% of maximum, it can be deduced that the antilipolytic effects of insulin and NMT were barely additional when acting in combination.

With respect to the stimulation of hexose transport, NMT, tyramine and synephrine, were incubated with a submaximal dose of insulin and human adipocytes, then 2-DG uptake was expressed as the percentage of the maximal response to 10 µM insulin (Figure 5). When present at 0.1 mM, none of these amines hampered the stimulation of 2-DG uptake by a low dose of insulin (10 nM), which averaged 75% of the maximal transport capacity. However, when human fat cells were incubated for 45 min with 10 nM insulin and 1 mM of each amine, they exhibited an impaired glucose transport stimulation (Figure 5). Thus, when co-incubated with insulin, NMT and related protoalkaloids were producing in vitro an apparent state of resistance to insulin.

## 4. Discussion

The in vitro experiments performed in this study show that, as several other dietary amines, NMT stimulates directly and acutely glucose uptake, at least when incubated at millimolar doses with functional human adipocytes. Such stimulation occurs in less than one hour. Although significant, this stimulation reached only 20–30% of the activation of glucose transport found in response to 100 nM insulin, which averaged a fourfold increase over baseline in all the subsequent adipocyte batches studied. Evidently, the transport capacity of functional human adipocytes is much lower than that found in rodent fat cells [28] but is comparable to that of preadipose cell lineages [29]. The moderate activation of glucose transport by NMT, approximately equivalent to a doubling of baseline, is qualified throughout this report as ‘insulin-like’ just to traduce its capacity to partially mimic a functional response to insulin, namely glucose uptake activation. It should be noted that such term ‘insulin-like’ does not mean that the entire insulin-signaling pathway was activated in response to the amines studied. A stimulation of glucose uptake in fat cells also occurred at a lower dose of NMT (0.01 mM), but to a lesser extent. Thus, since NMT effect was essentially observed at millimolar doses, one might suspect it to be non-specific. However, activation of 2-DG uptake was not observed for all amines tested (e.g., prenylamine and glucosamine). Moreover, since NMT effect was impaired by amine oxidase inhibitors, it can be proposed that it was MAO and/or SSAO dependent rather than artifactual. Although an interaction with membrane receptors could not be ruled out, we propose a mechanism of action that mainly relies on enzyme-mediated metabolism of NMT. Such paradigm is consistent with the high doses of NMT needed to achieve stimulation of glucose transport, which are closer to the K_m_ of MAO and SSAO for most of their substrates than to the K_D_ of TAAR-1 for its ligands [30]. Moreover, previous studies have demonstrated that the oxidation of tyramine, benzylamine or methylamine by MAO or SSAO triggers the translocation of glucose transporters to the plasma membrane in a manner prevented by various inhibitors and antioxidants [19,31,32]. Thus, NMT likely triggers glucose carrier recruitment at the cell surface via the inhibition by hydrogen peroxide of tyrosine phosphatases involved in the continuous vesicular trafficking and in the turn-off of insulin signaling [33]. Indeed, hydrogen peroxide is a common end-product of amine oxidation, irrespective of the substrate or the oxidase considered. Hydrogen peroxide is the sole oxidation product exhibiting insulin-like effects, since aldehydes or ammonia (i.e., the other end-products of amine oxidation) were not activating hexose uptake [34,35] or inhibiting lipolysis [36]. Furthermore, a similar insulin-like effect has been evidenced for various amines in other cell types that co-express high levels of the glucose carriers GLUT1 and GLUT4 together with MAO (case of cardiomyocytes) [37], or SSAO (case of smooth muscle cells) [32], or both amine oxidases (case of adipocytes) [19].

The current study shows for the first time the capacity of NMT to activate hexose transport in human fat cells, in a manner sensitive to MAO and SSAO inhibition. In fact, the insulin-like effect of NMT was apparently very close to that of its parent molecule tyramine, which has been demonstrated in human fat cells to depend predominantly on MAO-dependent hydrogen peroxide generation [12]. It must be kept in mind that activation of glucose transport is the first step of the assembly of triacylglycerols in fat cells and it could be likely supposed that amines activating such step facilitate de novo lipogenesis in hAT [38], as demonstrated in rat adipocytes [12].

Among the limitations of the current study is the fact that hydrogen peroxide production in response to NMT has not been directly evidenced in adipocytes. Moreover, the relative proportion of MAO and SSAO involved in the resulting insulin-like effect of NMT has not been sufficiently quantified even though experiments examined the direct inhibition of glucose uptake by four molecules with recognized inhibitory action on amine oxidase activities: pargyline [39], aminoguanidine [40], BTT 2052 [26] and BFI [41]. Even when studying the competition for the oxidation of [^14^C]-tyramine and [^14^C]-benzylamine in hAT homogenates, the relative proportion of MAO and/or SSAO involved in the metabolism of NMT could not be determined precisely. At least, our approaches indicated that NMT was interacting with human MAO in a manner similar to tyramine and serotonin, two well-described MAO substrates, thereby confirming the pioneering findings of Singer and coworkers, showing that NMT is a competitive substrate for rodent MAO [42]. NMT interacted with human SSAO only to a lesser extent, which could not be accurately quantified. Anyhow, higher precision about the nature of the amine oxidase involved in the observed short-term effects of NMT is not immediately necessary for delineating its mechanism of action since both MAO and SSAO substrates have been reported to similarly activate glucose uptake in human adipocytes, as it is the case for the reference MAO substrate tyramine [12] and for benzylamine, the reference SSAO substrate [24].

Both MAO and SSAO substrates have been reported to promote glucose uptake in rodent fat cells as well [37], alongside to amines such as adrenaline and noradrenaline, which were astonishingly active in an amine oxidase-independent manner [43]. Even when long-term actions of amines were considered, such as the promotion of adipocyte differentiation in cultured murine preadipocytes (which requires at least seven to eight days of treatment) both MAO and SSAO substrates were equivalent in mimicking or in reinforcing the adipogenic effect of insulin [44]. In view of such similarities between the insulin-like effects of various MAO and SSAO substrates, it is worthy to mention that a complete inhibition of the tyramine activation of hexose transport has been obtained in rat adipocytes only when both MAO and SSAO activities were blocked [19]. Tyramine is a substrate of both MAO and SSAO in rat [19], while it is only a MAO substrate in humans [12]. Thus, it is no so astonishing to observe here that 0.1 mM pargyline almost abolished both tyramine-induced hexose uptake and [^14^C]-tyramine oxidation. Accordingly, the capacity of the SSAO inhibitors to impair NMT and tyramine insulin-like effects in human adipocytes was likely related to a loss of their selectivity and an impairment of both MAO and SSAO activities when tested at 1 mM.

NMT also mimicked the antilipolytic effect of insulin in human adipocytes. Such observation was a confirmation of its capacity to inhibit glycerol and free fatty acid release by isolated adipocytes [17]. Since the potential blockade by MAO and SSAO inhibitors was already published in [17], this approach was not performed again. However, while the SSAO inhibitor BTT 2052 did not block the antilipolytic effect of NMT in this previous study, it impaired here its activation of glucose uptake. Such apparent discrepancy relies with the different doses of BTT 2052 used: 10 µM in the former and 1 mM in the present study. Again, a loss of selectivity for SSAO inhibition could be invoked for the high dose of BTT 2052.

Otherwise, it must be noted that NMT and the other related amines were more antilipolytic than insulin in human adipocytes. The inhibition of isoprenaline-stimulated lipolysis was the sole effect for which NMT and the reference amines reached a maximum that was higher than the response observed for 100 nM insulin. For this reason, the amines cannot be simply qualified as ‘partial’ insulin mimickers. A putative interplay between NMT and Gi-coupled membrane receptors, especially the α_2_-adrenergic receptors, highly antilipolytic in human fat cells [45], might constitute an explanation for such substantial antilipolytic action. In line with this, agonism at α_2_-adrenergic receptors has already been proposed for NMT in fat cells [16] and in other models [46]. Since the antilipolytic effect of NMT and related amines was larger in magnitude than that of insulin, it is this property that should deserve future studies, especially those dealing with the correction of lipotoxicity, a metabolic dysregulation of diabetes and obesity, for which dietary amines have recently been proposed as beneficial [47]. Though the activation of lipolysis by NMT was very partial and less attractive, further investigation on possible partial agonism/antagonism at β-adrenergic receptors remains of interest since this eventuality cannot be excluded, and might explain the partial lipolytic component of NMT and its impairement of isoprenaline effect.

Among the reference dietary amines used in this study was synephrine, the most abundant amine found in fruits of *Citrus* species [48], which differs from NMT by a single additional hydroxyl group (see the chemical structures in the graphical abstract or in freely accessible websites). Although synephrine lipolytic effect is advertised by non-scientific websites promoting the consumption of diet supplements, we have already reported a weaker lipolytic effect of this agent in human than in rodent fat cells [16]. However, this amine remained the most active of the partial lipolytic agents tested here and was not surpassed by NMT. As already observed in human fat cells [49], the partial insulin stimulation of hexose uptake by synephrine was confirmed here. Surprisingly, the current study showed that synephrine also impaired the insulin action on hexose uptake. Such complex interplay between synephrine and insulin effect on glucose disposal is much less recognized than its widely advertised lipolytic effect [50,51] or its demonstrated improvement of fat oxidation during exercise [52]. For synephrine and its parent NMT, the in vitro effects occurred at high doses (0.1–1 mM) and were complex since they modestly contributed to activate both the accumulation and the breakdown of triacylglycerols in human fat cells. Consequently, these two related biogenic amines cannot simply be advertised as potent lipolytic agents.

Octopamine is also a dietary amine related to NMT and synephrine, and which coexist in *Citrus* fruits [53] and in a few weight-lowering dietary supplements [54,55,56]. The literature indicates that octopamine stimulates 2-DG uptake in rodent [18] and human adipocytes [49]. It has been suggested that octopamine behaves as a MAO substrate in human fat cells since it competed for [^14^C]-tyramine oxidation [18]. However, octopamine has been considered more recently as a SSAO inhibitor since it does not generates hydrogen peroxide when incubated with the purified human form of the enzyme [57]. Even more complex is its capacity to activate the β_3_-adrenergic receptors and to be strongly lipolytic in rodent but not human adipocytes [54]. All these biological effects of octopamine are relatively close to those depicted here for NMT in human adipocytes. Nevertheless, NMT is much less lipolytic than octopamine in rodent adipocytes [16]. In fact, since synephrine is more abundant than octopamine, NMT or hordenine in orange fruits [13]. It should be the main responsible for the lipolytic action, if any, in humans, of *Citrus aurantium* extracts claimed to mitigate obesity. Nevertheless, the current work definitely states that isoprenaline (also known as isoproterenol), which is not present in *Citrus*, is clearly more lipolytic in human fat cells than any of the dietary amines found in *Citrus* fruits. It is worth mentioning that isoprenaline binds with high affinity to β-adrenergic receptors without activating other membrane receptors and without interacting with MAO or SSAO enzymes, as evidenced here by its low efficiency in competing for [^14^C]-tyramine and [^14^C]-benzylamine oxidation. Moreover, the strong lipolytic capacity of isoprenaline is not accompanied by any activation of glucose uptake in human fat cells [58] or sensitivity to deamination by MAO. Despite its selective and potent lipolytic property, isoprenaline is not among the currently used anti-obesity drugs (reviewed in [59]) due to its cardiovascular side-effects. Although less active on the β-adrenergic receptors [16], NMT, is also clearly less lipolytic than isoprenaline, and even endowed with an antilipolytic component, and a feeble capacity to favor glucose consumption by fat cells.

Taken as a whole, our observations do not support the use of NMT as anti-obesity agent. However, the actions of NMT reported in the current study were limited to a panel of short-term effects and do not exclude any beneficial long-term nutrigenomic action of NNT supplementation, which is difficult to investigate on the human cell type used here. This is another limitation of the present study, together with lack of demonstration of the in vivo relevance of the millimolar doses tested in vitro.

Lastly, another possible limitation of this study is omitting to test NMT in the presence of insulin, since the ingestion of NMT, present in food and beverages or consumed via dietary supplements, will lead to a concomitant increase in insulin and dietary amines in hAT during postprandial periods. When NMT and tyramine were tested in the presence of insulin, both impaired the stimulatory action of the pancreatic hormone rather than reinforcing its activation of glucose uptake. Such behavior of insulin mimickers that partially reproduce several of the insulin actions when tested alone, but which impair the responsiveness to insulin when combined with it, has been often observed, and is mentioned here by only quoting the examples of okadaic acid and phenylarsine oxide [60]. Thus, such a class of partial insulin mimickers is probably not beneficial for alleviating insulin resistance.

## 5. Conclusions

By highlighting the complex and weak effects that NMT exerts directly on human fat cells, our current investigations do not support its purported use to alleviate fat accumulation, as a substitute for banned ephedrine in dietary supplements marketed for fat mobilization. Of course, our results are limited to a cellular model and cannot exclude other peripheral or central effects. Nevertheless, our results are in agreement with a review of the pharmacological properties of NMT, leading to the conclusion that there is no valuable demonstration of its efficiency in improving performance or facilitating weight loss [61]. Whether future studies with a solid scientific basis will detect non-shivering thermogenesis activation or anorectic action for NMT administration in humans, it should be noted that, at present, only high doses of this amine weakly activate triacylglycerol turnover in human adipocytes, and that no direct effect has been detected at lower doses.

## Figures and Tables

**Figure 1 nutrients-14-03118-f001:**
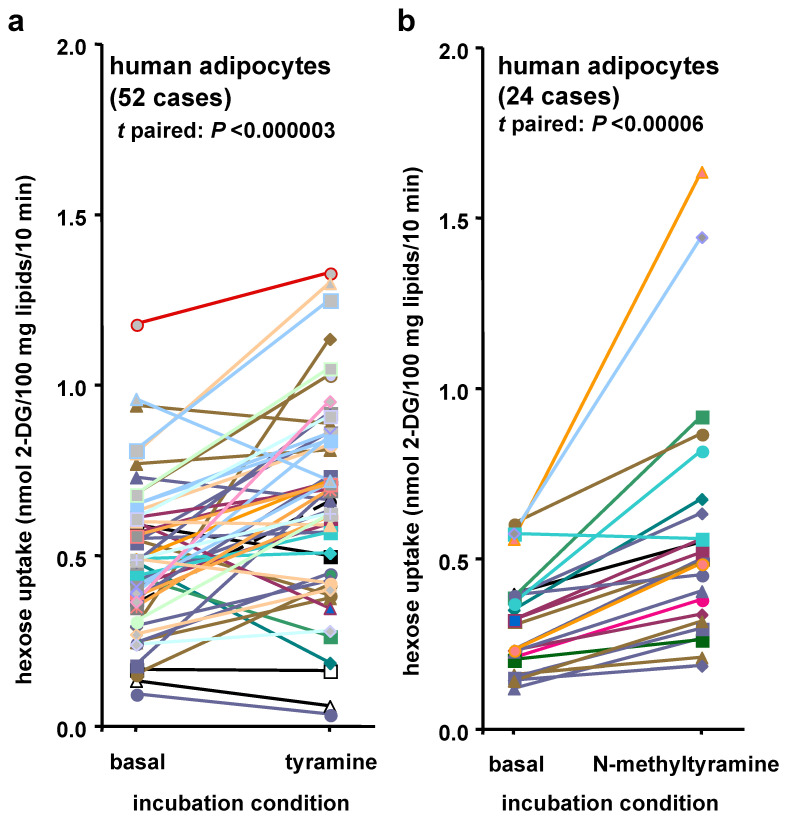
Influence of tyramine and *N*-metyltyramine (NMT) on hexose uptake in human adipocytes. (**a**) For each individual adipocyte preparation from a total of 52 subjects (one color per individual), the comparison is shown between 2-deoxyglucose uptake (2-DG) determined after 45 min incubation without (basal) or with 1 mM tyramine. Significant stimulatory influence of tyramine was validated by *t*-paired analysis (*p* < 0.000003). (**b**) Same *t*-paired analysis between spontaneous 2-DG uptake (basal) and in response to 1 mM NMT (*N*-methyltyramine) is shown for a total of 24 other individuals. Significant difference between basal and NMT-stimulated uptake was found at *p* < 0.0006.

**Figure 2 nutrients-14-03118-f002:**
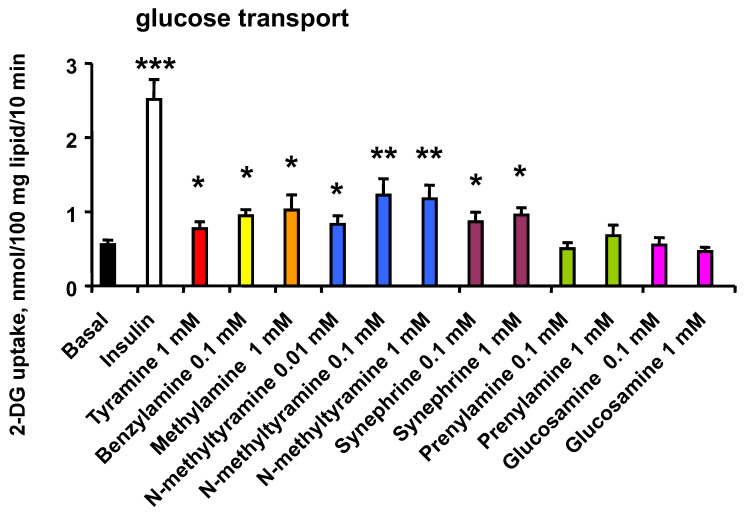
Activation of glucose transport in human adipocytes by insulin and several -but not all- tested amines. The uptake of 2-deoxyglucose (2-DG) was assessed on 10 min after 45-min incubation of human subcutaneous adipocytes without (basal) or with 100 nM insulin (open column), or with the indicated concentrations of: tyramine (red), benzylamine (yellow), methylamine (orange), NMT (blue columns), synephrine (purple columns), prenylamine (green columns) or glucosamine (pink columns). Each column is the mean ± SEM of 8 determinations (mean BMI of subjects was: 24.6 ± 1.3, age range: 34–56 year). 2-DG uptake was expressed as nmoles of hexose transported/100 mg cell lipids/10 min. The mean amount of cell lipids per assay was 20 ± 2 mg. Significant influence was evidenced by ANOVA followed by Dunnett’s test by difference from basal at: * *p* < 0.05; ** *p* < 0.01; *** *p* < 0.001.

**Figure 3 nutrients-14-03118-f003:**
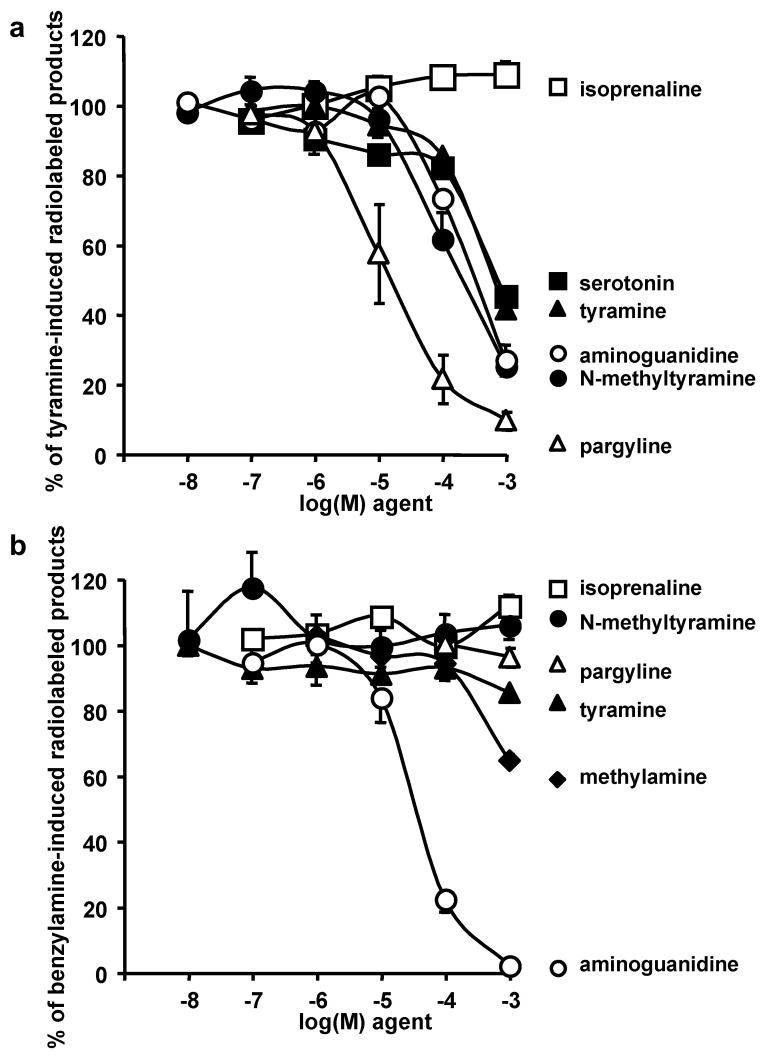
Influence of tyramine, NMT, and reference substrates or inhibitors on MAO and SSAO activities in human adipose tissue. (**a**) The oxidation of 0.5 mM [^14^C]-tyramine by the MAO activity found in human AT homogenates was set at 100% in the absence of any added agent. The release of radioactive products (mainly aldehydes) derived from [^14^C]-tyramine oxidation was almost abolished by the MAO inhibitor pargyline (open triangles), and inhibited by increasing doses of aminoguanidine (open circles), *N*-methyltyramine (NMT, closed circles), serotonin (closed squares), or cold tyramine (closed triangles), but not by isoprenaline (open squares). (**b**) [^14^C]-benzylamine oxidation was not inhibited by pargyline but was abolished by the SSAO inhibitor aminoguanidine. [^14^C]-benzylamine oxidation was not modified in the presence of isoprenaline, *N*-methyltyramine or tyramine, but was impaired by 1 mM methylamine (closed diamonds). Each point is the mean ± SEM from five to seven preparations of human subcutaneous adipose tissues. For each panel, no significant difference was found between tyramine and NMT.

**Figure 4 nutrients-14-03118-f004:**
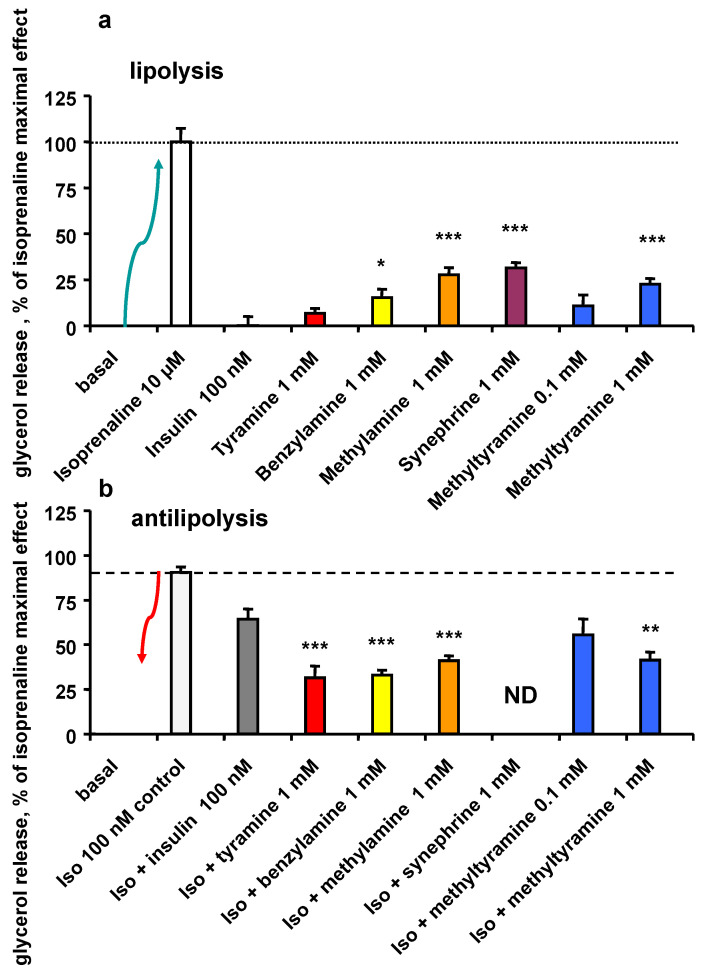
Influence of tyramine, NMT, and reference amines on lipolysis activation or inhibition in human adipose cells. Maximal glycerol release in response to 10 µM isoprenaline was set at 100% (open column), while basal lipolysis was set at 0%. Glycerol release in the presence of the indicated agents was expressed as the percentage of maximal lipolysis. (**a**) Lipolytic activation (green arrow) was partial and significantly lower than the maximal lipolytic effect of isoprenaline (dotted line) with all the tested amines. (**b**) Antilipolytic effect (red arrow) corresponds to the inhibition of glycerol release provoked by 100 nM isoprenaline (open column and dashed line). All tested amines significantly inhibited the isoprenaline-activated glycerol release: tyramine (red column), benzylamine (yellow column), methylamine (orange column) or NMT (methyltyramine, blue columns). Each column is the mean ± SEM of 14–16 determinations. For (**a**,**b**), significantly different from the effect of insulin (grey column) at: * *p* < 0.05; ** *p* < 0.01; *** *p* < 0.001. Iso: isoprenaline 100 nM. ND: not determined.

**Figure 5 nutrients-14-03118-f005:**
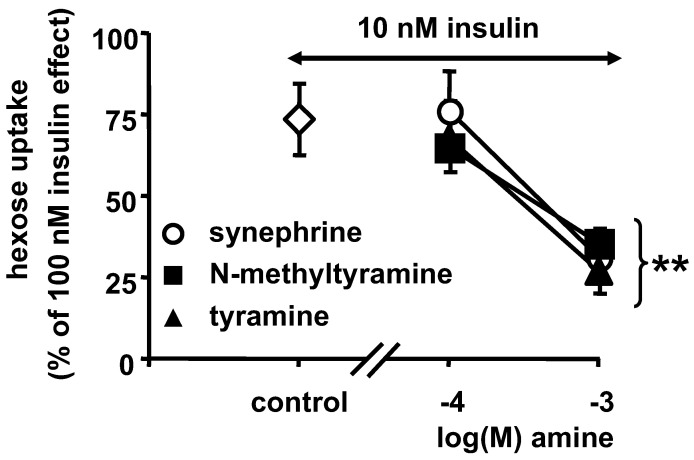
Influence of NMT, tyramine and synephrine on insulin-dependent stimulation of glucose transport in human adipocytes. The submaximal stimulation of hexose uptake by 10 nM insulin was expressed as the percentage of the maximal response to 100 nM insulin (set at 100%, with basal 2-DG uptake set at 0%). Insulin was tested alone (control, open diamond) and in the presence of the indicated concentrations of amines. Mean ± SEM of six preparations of human adipocytes. Significant difference from control at: ** *p* < 0.01.

**Table 1 nutrients-14-03118-t001:** Sensitivity of the amine-induced hexose transport to amine oxidase inhibitors in freshly isolated human adipocytes.

2-DG Uptake (% of Ins Max Effect)	Basal	Bza 1 mM		NMT 1 mM		Tyr 1 mM	
control	0	44.2 ± 9.9		33.1 ± 6.6		24.2 ± 4.6	*
+0.1 mM pargyline	1.3 ± 10.2	38.7 ± 5.3		20.2 ± 8.0		9.8 ± 2.0	*
+1 mM aminoguanidine	8.1 ± 6.1	7.7 ± 5.9	**	2.5 ± 6.8	**	6.9 ± 3.1	*
+1 mM BTT 2052	5.7 ± 6.3	6.2 ± 5.1	**	5.9 ± 6.3	*	8.7 ± 2.6	*
+0.1 mM BFI	11.2 ± 4.7	ND		6.2 ± 7.9	*	7.0 ± 5.0	*

2-deoxyglucose uptake (2-DG) was expressed as the percentage of maximal activation by insulin (% ins max effect), with 100 nM insulin set at 100% and basal uptake set at 0% in each of 8 separate preparations of human adipocytes. These percentages could pass below 0% when 2-DG uptake was lower than baseline (basal) and are shown as the means ± SEM of 6–8 determinations. Each column corresponds to the uptake measured after 45 min incubation without activator (basal) or in the presence of 1 mM of benzylamine (Bza), *N*-methyltyramine (NMT) or tyramine (Tyr). The amine oxidase inhibitors were added at time 0 at the indicated dose or absent (control, first line). Different from corresponding control condition at: * *p* < 0.05; ** *p* < 0.01. BFI: 2-benzofuranyl-2-imidazoline. BTT 2052: 2-(1-methylhydrazino)-1-indanol. ND: not determined.

## Data Availability

No additional data are available.

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
