# Peer review of "Multiple Direct Effects of the Dietary Protoalkaloid N-Methyltyramine in Human Adipocytes"

_nutrients, 2022, doi:10.3390/nu14153118_

Round 1

Reviewer 1 Report

The manuscript seeks to examine possible "insulin-like" effects of N-methyltyramine and several related amines in primary human adipocyte preparations. While I have no concerns about the main conclusions as to the usefulness (or lack thereof) of such amines as dietary weight loss supplements, there are several areas of the manuscript that I feel require attention. 

1) Reporting tyramine as "owing almost all its activity to the release of endogenous noradrenaline" (Introduction) is far too simplistic. While it is true that at high concentrations, far exceeding those seen in the body under normal conditions, tyramine has well established indirect sympathomimetic properties, it is well established that at orders of magnitude lower concentrations it activates TAAR1, and TAAR1 has very well defined physiological functions.

2) The introduction is almost devoid of known TAAR1 mediated effects on glucose and fat utilization, insulin (and other nutrient-related hormone) secretion, food intake, and weight loss. While the authors correctly identify in the discussion that the concentrations they have used far exceed the Kd of the amines at TAAR1, for completeness when discussing such effects of tyramine.

3)  A central conclusion is that effects of these amines are actually hydrogen peroxide mediated due to amine oxidase metabolism. While amine oxidase inhibitors were shown to prevent the effects of the amines, this reviewer found it puzzling that there was no increased response in the 2DG uptake assay following 1 mM NMT. Presumably 1mM NMT/synephrine produces more H2O2 than 0.1mM. Are synephrine/prenylamine amine oxidase substrates? 

4) There appears to have been no statistics performed on the data in Figure 4 (both panels) Specifically strong claims are made of differences in the effects of amines compared to insulin. Supportive significance needs to be shown, or in its absence a fairly considerable re-write of the relevant sections. 

5)  With respect to effects observed in the presence of insulin (Figure 5 and the preceding text reported results), 0.1 mM NMT has maximal effects alone (not increased at 1mM). How can the effects of 0.1mM NMT not be additive to insulin if is mediated by H2O2 from NMT metabolism? 

6) If the mechanism is through H2O2 turning off insulin signalling (discussion) why do the authors see effects of NMT alone? How much "constitutive" insulin signalling is occurring in these preparations?

Author Response

Point-by-point answers to reviewers:

Reviewer #1:

The manuscript seeks to examine possible "insulin-like" effects of N-methyltyramine and several related amines in primary human adipocyte preparations. While I have no concerns about the main conclusions as to the usefulness (or lack thereof) of such amines as dietary weight loss supplements, there are several areas of the manuscript that I feel require attention.

            Thank you for your careful reading. We agree with referee's remarks, and we hope that the inputs made during revision will improve our message. The modifications appear in red in the submitted revision.

1) Reporting tyramine as "owing almost all its activity to the release of endogenous noradrenaline" (Introduction) is far too simplistic. While it is true that at high concentrations, far exceeding those seen in the body under normal conditions, tyramine has well established indirect sympathomimetic properties, it is well established that at orders of magnitude lower concentrations it activates TAAR1, and TAAR1 has very well defined physiological functions.

            Thank you for your comment. We agree with referee's remarks, but we wanted to introduce tyramine in an 'historical' manner, mentioning first its catecholamine releasing actions known for decades prior the discovery of TAARs. Please, note that the two halves of tyramine action were in the same introductive sentence. Authors could not improve with succinity such introduction. Nevertheless, the detailed review on TAARs and their ligands published by Gainetdinov, Hoener and Berry in 2018 has been added to the references quoted at the end of this introductive sentence.

2) The introduction is almost devoid of known TAAR1 mediated effects on glucose and fat utilization, insulin (and other nutrient-related hormone) secretion, food intake, and weight loss. While the authors correctly identify in the discussion that the concentrations they have used far exceed the Kd of the amines at TAAR1, for completeness when discussing such effects of tyramine.

            Thank you for this suggestion. We have added a sentence and two references (Raab et al.; Moore et al.) dealing with the effects of TAAR1 activation on gastrointestinal and pancreatic islet hormone secretion and eating behaviour, hoping that these changes will satisfy the query for introducing the promising roles of TAAR1 for the treatment of type 2 diabetes and obesity.

3) A central conclusion is that effects of these amines are actually hydrogen peroxide mediated due to amine oxidase metabolism. While amine oxidase inhibitors were shown to prevent the effects of the amines, this reviewer found it puzzling that there was no increased response in the 2DG uptake assay following 1 mM NMT. Presumably 1mM NMT/synephrine produces more H2O2 than 0.1mM. Are synephrine/prenylamine amine oxidase substrates?

            Thank you for these clever comments. The reviewer is absolutely right in being surprised by he lack of dose-dependency increase in the activation of glucose uptake by N-methyltyramine and synephrine in Figure 2. We were aware of this since it was commented in Results that " An increase of glucose uptake was also detected with 0.01 mM NMT, while a plateau was reached between 0.1 and 1 mM NMT (Figure 2)". Thus, this key sentence was not modified. Regarding the amine oxidase substrate nature of N-methyltyramine, we are unable to provide more data than shown in Fig. 3 for the competition for tyramine and benzylamine oxidation. Since we have not demonstration of the dose-dependent H2O2 release in response to NMT as presumed by the referee, it was added in Discussion the following sentence: " Among the limitations of the current study is the fact that hydrogen peroxide production in response to NMT has not been directly evidenced in adipocytes.". Nevertheless, other sentences underlined that the effects of NMT on lipid accumulation/mobilization were the primary objective of this work, and that there were characterized. As told above by the reviewer for tyramine, this compound and relatives not only activate TAAR1, they can also interact with many other targets belonging to the physiology of monoaminergic neurotransmission: transporters, receptors, enzymes involved in the synthesis or degradation of such amines. It is therefore no so astonishing to found that N-methyltyramine and synephrine could act as partial agonists/antagonists on beta- or alpha-adrenergic receptors (ARs), or other G-coupled proteins without producing classical sigmoidal dose-response curves. Thereby, they poorly stimulate lipolysis at 1 mM only (see Figure 4). This possibility was just evoked in Discussion since no experimental demonstration was sustaining this hypothesis. Thus, the effect of 1 mM N-methyltyramine on adipocytes (triacylglycerol synthesis or breakdown) is the resulting balance between these complex interactions, resulting in the appearance of biphasic dose-response curves or presence of plateau as shown here for 2-DG uptake. Can it be reminded here that norepinephrine (noradrenaline) itself exhibit a biphasic dose-reponse curve on lipolysis in human adipocytes (see Carpéné et al. Pharmaceuticals; doi 10.3390/ph13030041)? Finally, it was also clearly indicated in the third paragraph of Discussion that the effect of high doses of N-methyltyramine and reference agents on lipolysis/antilipolysis have been more deeply studied in rodent adipocytes in a previous report, quoted as ref 14 (Leroux et al., 2019).

            Unfortunately, we did not find in our files any data about influence of synephrine and prenylamine on amine oxidase activity either as substrate or blocker, with the exception that they were far from being among the best molecules interacting with SSAO activity in a high-throughput screening: synephrine and prenylamine leading respectively to 33 and 26% inhibition of benzylamine oxidation, while the best inhibitor phenelzine reached 98% inhibition. This is consistent with the lack of SSAO substrate activity we presumed for these amines, used as 'negative' control.

4) There appears to have been no statistics performed on the data in Figure 4 (both panels) Specifically strong claims are made of differences in the effects of amines compared to insulin. Supportive significance needs to be shown, or in its absence a fairly considerable re-write of the relevant sections.

            We respectfully disagree with your comments, since in the original version of the Ms, it was specified at two occurrences in the legend that " Lipolytic activation (green arrow) was partial and significantly lower than the maximal lipolytic effect of isoprenaline (dotted line) with all the tested amines. " and that " All tested amines significantly inhibited the isoprenaline-activated glycerol release ". Therefore, statistical analyses were performed, but no asterisk was included in the figure 4 to avoid the appearance of a useless "constellation of stars". These sentences in legend were not removed because the reference for lipolysis remains isoprenaline responsiveness in authors' opinion. However, since the reviewer was right to remind that insulin is the gold standard 'reference' for the tested amines, we have re-performed other statistical analyses for comparison between insulin and amines, and indicated the resulting significant differences as symbols added in novel figure 4. Thanks for helping us in improving the understanding of our figures in accordance with our interpretations.

5) With respect to effects observed in the presence of insulin (Figure 5 and the preceding text reported results), 0.1 mM NMT has maximal effects alone (not increased at 1mM). How can the effects of 0.1mM NMT not be additive to insulin if is mediated by H2O2 from NMT metabolism?

            This comment is very difficult to be taken into account, since all co-authors agree with the somewhat unexpected nature of results as noted by the reviewer. However, when experiments give always the expected results, one is no more at the level of exploratory investigation but only at that of practical curses for students at University! The reviewer has well understood how embarrassing were the obtained data with the combination of insulin and NMT, especially in the case of glucose transport stimulation. In the Discussion, we commented such situation by quoting several of the numerous reports indicating that most of the so-called 'insulin-like' agents are capable to activate glucose entry in target cells in the absence of insulin only. Most of them impair the action of the pancreatic hormone when used in combination with it: okadaic acid, phenylarsine oxide, phorbol esters... It is obviously the case of NMT here at 0.1 and 1 mM. It is therefore no so astonishing to observe that NMT was not additive to insulin at the lower dose , since it hampered the activation of glucose uptake by insulin itself when tested at the higher dose. A limiting step was likely reached with such huge amount of NMT. even if behaving only as an amine oxidase substrate, the amount of H2O2 supposed to be released with 1 mM NMT is probably out of the range of its intracellular messenger function. In this case, the endogenous antioxidant defences were overpassed and an oxidative stress might be generated. On the one hand, there is inhibition of the enzymes involved in the turn off of the cascade of tyrosine phosphorylation downstream insulin receptor activation, while on the other hand there is also an impairment of elements required for the translocation of glucose transporters and their proper function at the cell surface level. The situation could be even more complicated if one considers that NMT is loosing its specificity as amine oxidase substrate at high doses and acts as a partial agonist or antagonist at membrane receptors, or even as an allosteric modulator of other enzymes. Let the authors simply return to the original aim of the present study: is NMT really a potent activator of lipid metabolism in human adipose tissue, as advertised in non-scientifically based websites? The answer is likely no, since high doses of this natural molecule impair both the isoprenaline activation of lipolysis and the insulin activation of lipogenesis. However, this nutrient is able to modulate adipocyte functions in a rapid manner, which is detectable in vitro, but which seems of lower magnitude than its promised 'fat burning ' high capacity.  

6) If the mechanism is through H2O2 turning off insulin signalling (discussion) why do the authors see effects of NMT alone? How much "constitutive" insulin signalling is occurring in these preparations?

            Thank you for your remark. Vanadium, which is an historical reference among the phosphatase inhibitors, is able alone to activate hexose uptake in many cell types sensitive to insulin. In fact, there is a continuous cycling of the GLUT4 from intravesicular vesicles to the cell surface, even under basal conditions, in which most of these transporters are retained hidden in intracellular compartments, and the endogenous activity of tyrosine phosphatases is never nil, even in the absence of insulin. There is no need for a constitutive activation of insulin receptor and its downstream signalling as proposed by the referee. Thus, strong inhibition of the 'turn-off ' part of this cycle is just less potent as a direct activation of the turn-on process to accumulate glucose carriers at the cell surface, even in the absence of insulin. The phosphatases sensitive to vanadium are also sensitive to their redox state and can be transiently inhibited by endogenously formed or exogenously applied H2O2. We propose the latter event as a mechanism belonging to the influence of amine oxidase substrates, especially in adipocytes.   Please note that, consistent with this proposed mechanism, the AO substrates led to only one-third of the maximal glucose uptake activation seen with insulin. It has been found also by others to a similar extend in cell types such as smooth muscles (Mercier et al. Regulation of semicarbazide-sensitive amine oxidase expression by tumor necrosis factor-alpha in adipocytes: functional consequences on glucose transport. J Pharmacol Exp Ther 2003, 304, 1197-1208) or cardiomyocytes (Fischer et al. Contraction-independent effects of catecholamines on glucose transport in isolated rat cardiomyocytes. Am J Physiol 1996, 270, C1204-1210).

Reviewer 2 Report

I find the present manuscript written by Carpene C et al. to be extermely intersting for the scientific field. Moreover, the article is very well written, documented and the experiments performed are very well described. Therefore, I recommend its publication.

Here are some suggestion for improvement:

- Some parts of the manuscript require English editing (e.g., Lines 81 - 94). Please have the entire manuscript checked by a native speaker.

- Line 74: please remove "(MAO" as you already mentioned the abbreviation in Line 58

- Line 284: please correct the spelling of "benzylamine"

- Line 341: the same for "lipolytic"

- The Discussion part is a bit too long and difficult to read, but also very detailed.

- As several abbreviations were used within the text, please add an "Abbreviation list" at the end of the manuscript (as it will be helpful for the reader)

Author Response

Point-by-point answers to reviewers:

Reviewer #2:

I find the present manuscript written by Carpene C et al. to be extermely intersting for the scientific field. Moreover, the article is very well written, documented and the experiments performed are very well described. Therefore, I recommend its publication.

            Thank you for your careful perusal, your understanding and your enthusiasm.

Here are some suggestion for improvement:

- Some parts of the manuscript require English editing (e.g., Lines 81 - 94). Please have the entire manuscript checked by a native speaker.

            Sorry for the inconvenience, we did not realize how clumsy was this passage, and tried to improve its readability.

- Line 74: please remove "(MAO" as you already mentioned the abbreviation in Line 58

            OK, corrected

- Line 284: please correct the spelling of "benzylamine"

            OK, corrected

- Line 341: the same for "lipolytic"

            OK, corrected

- The Discussion part is a bit too long and difficult to read, but also very detailed.

            We agree with reviewer's comment, but for answering to the other reviewer, we have added several sentences.

- As several abbreviations were used within the text, please add an "Abbreviation list" at the end of the manuscript (as it will be helpful for the reader)

            OK, corrected
